# Purification of Functional Human TRP Channels Recombinantly Produced in Yeast

**DOI:** 10.3390/cells8020148

**Published:** 2019-02-11

**Authors:** Liying Zhang, Kaituo Wang, Dan Arne Klaerke, Kirstine Calloe, Lillian Lowrey, Per Amstrup Pedersen, Pontus Gourdon, Kamil Gotfryd

**Affiliations:** 1Department of Biomedical Sciences, University of Copenhagen, Nørre Allé 14, DK-2200 Copenhagen N, Denmark; liying.zhang@sund.ku.dk (L.Z.); kaituo@sund.ku.dk (K.W.); 2Department of Veterinary and Animal Sciences, University of Copenhagen, Dyrlægevej 100, DK-1870 Frederiksberg C, Denmark; dk@sund.ku.dk (D.A.K.); kirstinec@sund.ku.dk (K.C.); lillianclowrey@gmail.com (L.L.); 3Department of Biology, University of Copenhagen, Universitetsparken 13, DK-2100 Copenhagen OE, Denmark; papedersen@bio.ku.dk

**Keywords:** ion channels, overproduction, production platform, protein purification, *Saccharomyces cerevisiae*, sensors, transient receptor potential (TRP) channels, yeast

## Abstract

(1) Background: Human transient receptor potential (TRP) channels constitute a large family of ion-conducting membrane proteins that allow the sensation of environmental cues. As the dysfunction of TRP channels contributes to the pathogenesis of many widespread diseases, including cardiac disorders, these proteins also represent important pharmacological targets. TRP channels are typically produced using expensive and laborious mammalian or insect cell-based systems. (2) Methods: We demonstrate an alternative platform exploiting the yeast *Saccharomyces cerevisiae* capable of delivering high yields of functional human TRP channels. We produce 11 full-length human TRP members originating from four different subfamilies, purify a selected subset of these to a high homogeneity and confirm retained functionality using TRPM8 as a model target. (3) Results: Our findings demonstrate the potential of the described production system for future functional, structural and pharmacological studies of human TRP channels.

## 1. Introduction

Transient receptor potential (TRP) channels constitute one of the largest families of ion channels and serve as cellular sensors that permeate cations, such as calcium, magnesium and sodium, in response to physical or chemical stimuli. Human TRP channels are widely expressed throughout the body, including brain, heart, liver, lung, small and large intestine, skeletal muscle, skin, pancreas, as well as in inflammatory and immune cells [1,2]. Changes of temperature, pH, the concentration of chemicals or membrane potential modulate TRP channel activity, highlighting their essential physiological roles in the sensation of thermal shifts [3], pain [4], taste [5] and pressure [6]. Malfunction of TRP channels significantly contributes to the development of many pathological conditions, including bipolar disorder, diabetes mellitus, various types of cancer, coronary heart disease and muscular dystrophia [7]. Consequently, TRP channels represent attractive pharmacological targets and indeed several TRP-modulating compounds currently undergoing clinical trials [8].

Based on sequence homology, topology and function, human TRP channels are divided into six subfamilies (Figure 1A): TRPC (canonical), TRPV (vanilloid), TRPM (melastatin), TRPA (ankyrin), TRPP (polycystin) and TRPML (mucolipin). Moreover, two additional subfamilies, i.e., TRPN and TRPY have been identified in *Drosophila melanogaster* and yeast, respectively. Structurally, TRP channels assemble as homo- or hetero-tetramers with a single pore formed in the center. Each subunit consists of intracellular N- and C-termini, six transmembrane helices (TM1-TM6) interconnected by relatively short loops, with a pore forming loop (P) inserted between the TM5 and TM6 (Figure 1B,C) [9,10]. Currently, several structures are available for the TRP family (Table 1), including TRPA [11], TRPPP2-3 [12,13], TRPV1-6 [14,15,16,17,18,19], TRPC3-6 [20,21,22,23,24], TRPML1 and 3 [25,26], as well as TRPM2, 4 [27,28] and 7-8 [29,30] from different species. In addition to revealing the overall architectures, the gathered structural information provided mechanistic insights explaining fundamental regulatory and functional mechanisms [9,31], as well as facilitated drug development, with, e.g., TRPV1 being a highly medically relevant target [32].

Being membrane proteins, many types of studies of TRP channels are nevertheless hindered by the difficulty of producing protein samples of sufficient quantity and quality in an economically sustainable manner. Systematic analysis of heterologous expression systems utilized for production of TRP channels for structural studies reveals that mammalian and insect cell-based platforms are the most commonly used (Table 1). TRPV1 [14], TRPM2 [27] and TRPV3 [16] channels were isolated from modified human HEK cells, whereas TRPML3 was overproduced using an insect (*Sf9*) cell-based platform [26]. To our knowledge, yeast, i.e., *Saccharomyces cerevisiae*, has been exploited to deliver the structures of two TRP channels only, namely TRPV2 from *Rattus norvegicus* [15] and TRPV5 from *Oryctolagus cuniculus* [18], whereas no structure is available originating from a bacterial host, despite attempts [33].

For many researchers, the primary expression system, also for the production of integral membrane proteins, has traditionally been *Escherichia coli*, due to the ease of genetic manipulation, availability of optimized expression plasmids, high-speed of growth and low cost [34,35]. However, generation of many proteins from higher sources frequently requires a eukaryotic expression system, such as insect, mammalian or yeast as hosts [36]. Compared to *E. coli*, the establishment of insect or mammalian cell-derived expression is nevertheless cost- and time-consuming. In this context, *S. cerevisiae* has advantages as it offers a cheap and robust large-scale production of properly-folded proteins with post-translational modifications combined with user-friendly genetic manipulations and simple culture conditions [36,37,38]. Hence, yeast represents an attractive complement for synthesis of high-quality protein, which has potential to permit in-depth biophysical and biochemical characterization, as well as drug discovery of many important targets, including TRP channels, for basic and applied sciences.

Here, we describe the development of an economic and effective method to isolate purified, functional human TRP channels applying a previously described robust *S. cerevisiae* membrane protein production platform [34,36,37]. Briefly, we approached 11 selected human TRP members belonging to 4 different subfamilies and produced these as full-length channels C-terminally fused to green fluorescent protein (GFP). We proceeded further with one member from each subfamily, i.e., TRPC4, TRPV3, TRPML2 and TRPM8, screened for suitable detergents for membrane extraction and assessed the quality of the solubilized samples by florescence-detection size-exclusion chromatography (F-SEC). Subsequently, we performed large-scale purification using affinity chromatography and investigated homogeneity of the samples employing SEC. Finally, for TRPM8, a medically significant target for the development of drugs to treat cold-associated respiratory disorders [39] and prostate cancer, respectively [40], we confirmed retained channel function following reconstitution into artificial lipid bilayers. Overall, our results suggest that *S. cerevisiae* is suitable for obtaining large-scales of active human TRP channels for numerous down-stream applications. 

## 2. Materials and Methods

### 2.1. Cloning and Construction of Plasmids

All cDNAs encoding full-length human TRP channels were codon-optimized for *S. cerevisiae* and purchased from GenScript (New Jersey, NJ, USA). Codon-optimization algorithm involves adjustment of a variety of parameters, including codon adaptability, mRNA structure, and various cis-elements critical in transcription and translation. TRP channel cDNAs and GFP were PCR amplified with AccuPol DNA polymerase (Amplicon, Odense, Denmark) and the primers listed in Appendix A. Each TRP channel-GFP-His8 fusion was generated by homologous recombination by co-transforming the *S. cerevisiae* PAP1500 strain [41] with a TRP PCR fragment, a GFP PCR fragment, as well as BamHI, HindIII and SalI digested pEMBLyex4 vector [42]. The PCR primers were designed to encode a Tobacco Etch Virus (TEV) protease cleavage site (GENLYFQ↓SQF) between the TRP channel and the GFP-octa-histidine (His8)-tag. Transformants were selected on agar plates containing synthetic defined (SD) medium with leucine (60 mg L^−1^) and lysine (30 mg L^−1^). The accuracy of all constructs was confirmed by DNA sequencing.

### 2.2. Small-Scale Expression of TRP Channels and Live Cell Bioimaging

All TEV-GFP-His8-fusions were expressed in the *S. cerevisiae* strain PAP1500 essentially as previously described [43]. Briefly, transformants were inoculated in 5 mL of SD media [44] containing glucose (20 g L^−1^), leucine (60 mg L^−1^) and lysine (30 mg L^−1^), and grown overnight at 30 °C. The following day, 200 µL of the culture was transferred to 5 mL of the same media lacking leucine and grown for 24 h at 30 °C to increase plasmid copy number upon leucine deprivation. Subsequently, 5 mL of the culture was scaled up to 50 mL in the same medium for another 24 h and used to inoculate 2 L of media supplemented with amino acid (alanine (20 mg L^−1^), arginine (20 mg L^−1^), aspartic acid (100 mg L^−1^), cysteine (20 mg L^−1^), glutamic acid (100 mg L^−1^), histidine (20 mg L^−1^), lysine (30 mg L^−1^), methionine (20 mg L^−1^), phenylalanine 50 mg L^−1^), proline (20 mg L^−1^), serine (375 mg L^−1^), threonine (200 mg L^−1^), tryptophan (20 mg L^−1^), tyrosine (30 mg L^−1^) and valine (150 mg L^−1^)), glucose (10 g L^−1^) and glycerol (3% *v*/*v*). Following glucose consumption, protein expression was induced by adding galactose to a final concentration of 2% and allowed for 48 h at 15 °C until the cells were harvested. Obtained material typically yielded in ~10 g of wet cell pellet.

Localization of expressed TEV-GFP-His8-fused TRP channels was performed by bioimaging of GFP fluorescence in vivo using the Nikon Eclipse E600 microscope (Nikon, Japan) equipped with a Optronics MagnaFire camera (Optronics, Muskogee, OK, USA).

### 2.3. Membrane Preparation, Detergent Screens and F-SEC 

*S. cerevisiae* cells were homogenized mechanically (BioSpec, Bartlesville, OK, USA) using glass beads and crude membranes were prepared as previously described [45]. Briefly, following ultracentrifugation (205,000× *g*, 3 h, 4 °C), membranes were resuspended in ice-cold solubilization buffer (SB; 20 mM Tris-NaOH pH 7.5, 500 M NaCl, 10% glycerol, 1 mM EDTA, 1 mM EGTA) supplemented with SIGMAFAST protease inhibitor cocktail (Sigma, St. Louis Missouri, MO, USA), 1 mM PMSF (Sigma) and 2 mM 2-mercaptoethanol (Sigma), and stored at −80 °C until further use. Isolated membranes from yeast overexpressing the respective TRP channel were subjected to detergent screening to test solubilization efficacy. Briefly, solubilization was performed at vigorous rotation (2 h, 4 °C) in SB in the presence of n-dodecyl-D-maltoside (DDM; Anatrace, Maumee, OH, USA), n-decyl-D-maltoside (DM; Anatrace) or 2,2-didecylpropane-1,3-bis-β-D-maltopyranoside (LMNG; Anatrace) in a final concentration of 2% (1:1 detergent-to-membrane mass ratio) or n-dodecylphosphocholine (FC-12; Anatrace) or n-heksadecylphosphocholine (FC-16; Anatrace) in a final concentration of 1% (1:2 detergent-to-membrane mass ratio). Insoluble material was removed by ultracentrifugation (50,000× *g*, 20 min, 4 °C) and 20 µL of the supernatant was used to measure GFP fluorescence (excitation 485 nm, emission 520 nm) to estimate detergent extraction efficacy. The remaining material was loaded on Superose 6 HR 10/30 column (GE Healthcare, Copenhagen, Denmark) equilibrated with buffer composed of 20 mM Tris-NaOH pH 7.5, 150 mM NaCl and 0.03% DDM, and subjected to F-SEC performed on ÄKTA Pure system (GE Healthcare) equipped with a Prominence RF-20A fluorescence detector (Shimadzu, Kyoto, Japan).

### 2.4. Large-Scale Protein Production, TEV Protease Cleavage and SEC 

Large-scale expression of selected TRP channels was performed in 15-L bioreactors essentially as previously reported [42]. Briefly, 50 mL of the yeast culture described above was scaled up to 1 L in the same medium and grown overnight at 30 °C. The following day, the culture was used to inoculate 10 L of identical medium supplemented with 3% glucose, 3% glycerol, amino acids (excluding leucine), inorganic salts and vitamins, and the growth was performed in Applikon fermenters connected to an ADI 1030 Bio Controller (Applikon Biotechnology, Delft, Netherlands) with the automated maintenance of culture pH at 6.0. Approximately 18 h after inoculation, 1 L of 20% glucose was added to further increase the growth of cells. Following glucose consumption, protein expression was induced by adding galactose to a final concentration of 2% and allowed for 96 h at 15 °C until the cells were harvested. Obtained material typically yielded in ~150–200 g of wet cell pellet.

Large-scale protein purification was performed using immobilized metal ion affinity chromatography (IMAC) with crude membranes isolated from 40 g of fermenter-grown yeast cells (obtained from ~3 L of cell culture). Membranes were solubilized by vigorous rotation (3 h, 4 °C) in SB containing the detergent of interest in a final concentration of 2% (DDM or DM) or 1% (FC-16). Insoluble material was removed by centrifugation (35,000× *g*, 1 h, 4 °C) and the supernatant was loaded onto a HisTrap HP column (GE Healthcare), washed with 50 mL of IMAC buffer (20 mM Tris-NaOH pH 7.5, 500 mM NaCl, 10% glycerol and 3 × CMC of the respective detergent). Subsequently, bound protein was eluted in IMAC buffer supplemented with step-wise (60, 250 and 500 mM) imidazole gradient. Top IMAC fractions were pooled, concentrated on Vivaspin concentrators (MWCO 100 kDa; Sartorius, Göttingen, Germany) and the GFP-His8-tag was cleaved with TEV-His10-tagged protease (home source; 16 h, 4 °C) mixed with the protein sample in a ratio of 1:10 (*w*/*w*). Treatment with TEV protease was performed with concomitant dialysis against IMAC buffer supplemented with 20 mM imidazole performed in MWCO 10 kDa dialysis bags (ThermoScientific, Waltham, MA, USA). Following cleavage with TEV protease, reverse IMAC (RIMAC) was performed to rebind un-cleaved TEV-GFP-His8-fusions, free GFP-His8-tag and TEV-His10-tagged protease. Briefly, samples were loaded onto a HisTrap HP column (GE Healthcare) equilibrated with 15 mL of IMAC buffer containing 20 mM imidazole, and the flow through was collected and concentrated to ~5 mg mL^−1^ as already described. Subsequently, RIMAC- or IMAC-pure samples were loaded onto a Superose 6 HR 10/30 column (GE Healthcare) equilibrated with SEC buffer (20 mM Tris-NaOH pH 7.5, 150 mM NaCl, 10% glycerol and 3 × CMC of the respective detergent). Following each SEC run, fractions corresponding to the main elution peak were pooled and concentrated to ~1 mg mL^−1^ as already described.

### 2.5. Measurements of Single Channel Ion Conductance

Single channel ion conductance was measured in lipid bilayers using an Orbit Mini workstation (Nanion Technologies, München, Germany) with 50 μm MECA4 recording chips with 4 microelectrode cavities (Ionera Technologies, Freiburg, Germany) as previously described [46]. Voltage was controlled by the EDR 3 software (Elements, Cesena, Italy). 150 μL of the recording solution (10 mM HEPES pH 7.2, 200 mM KCl, 100 mM NaCl and 0.2 mM CaCl_2_) was added to the cavity of the chip and electrical contact was established between the electrodes. Lipid planar membranes were formed over the electrode using 10 mM of 1,2-diphytanoyl-sn-glycero-3-phosphocholine (DPhPc; Avanti Polar Lipids, Alabaster, AL, USA) and 1 mM cholesterol in n-nonane (Avanti Polar Lipids). DDM-solubilized SEC-purified TRPM8 sample (0.2 µL of 6.34 µg µL^−1^) was added to recording solution close to the bilayers on the cis-side of the chip. Voltage was switched between positive and negative values, increasing by 20-mV intervals to stimulate protein insertion into the membrane. To activate TRPM8, 1.0 μL of icilin (Sigma) was added to a final concentration of 0.2 μM. Once channel activity was detected, voltage steps to −20, −40, −80, −100, 20, 40, 60, 80 and 100 mV were applied. If no activity was observed after more than 2.5 min, the solution was gently mixed, and if still no activity, an additional 0.2 μL of protein sample was added. All recordings were obtained at room temperature (20–22 °C). 

Recordings were low-pass filtered at 0.5 kHz and analyzed using Clampfit 10.7 software (Molecular Devices, San Jose, CA, USA). For single channel events current amplitude was determined. Single channel current was plotted as a function of voltage (I/V curve) using Prism 7 software (GraphPad, San Diego, CA, USA) and slope conductance was determined by linear regression. 

## 3. Results

### 3.1. Selected TRP-Channel Targets and The Expression System

To investigate the capacity of *S. cerevisiae* to serve as a host for production of human TRP channels, we selected 11 human members belonging to 4 of the main subfamilies, hence, representing targets with a variety of distinct structural (e.g., N- and C-termini) and functional features. Specifically, the portfolio included the following members: TRPC3, TRPC4, TRPC5, TRPV1, TRPV3, TRPV4, TRPML1, TRPML2, TRPML3, TRPM1 and TRPM8. Each expression plasmid encompassed full-length sequences codon-optimized for *S. cerevisiae* with C-terminal TEV-GFP-His8-tag fusions (Figure 2A) to enable visualization, affinity purification and quality control of generated targets. Constructs were designed to employ the hybrid promoter (CG-P) of the pEMBLyex4 vector, enhanced by the PAP1500 yeast production strain overexpressing the GAL4p transcriptional activator (Figure 2B) [42]. The potency of the system is further increased through selection for leucine autotrophy that significantly increases the plasmid copy number prior to induction of the protein of interest [47].

### 3.2. Small-Scale Production and Localization in S. cerevisiae

As low protein yields obstruct most types of downstream studies, the obtained quantity of the targets is critical to evaluate. Taking advantage of GFP detection of target accumulation permitted by the fusion tag, we determined the produced levels of the 11 human TRP channels following 2-day induction in 2-L cultures. By measuring whole-cell GFP fluorescence [48] and a GFP standard [49], we estimated the target levels to range from 2.4 to 5.2 mg of protein per liter cell culture (Figure 3A), representing rather promising yields for further optimization to produce samples for biophysical, structural and drug discovery efforts. The highest protein levels were achieved for TRPC4, whereas TRPML1 expressed the lowest, but still with encouraging yields.

Subsequently, we took advantage of yet another useful feature of correctly-folded GFP, i.e., its SDS-resistance, and employed in-gel GFP fluorescence (visible following SDS-PAGE), to visualize the targets in crude membrane preparations (Figure 3B). Encouragingly, all the assessed 11 human TRP channels accumulated in the membranes in the full-length form. The electrophoretic mobility correlated well with the predicted molecular weight, with TRPM1 representing the heaviest target (wild-type MW of 198.7 kDa) and TRPML3 the lightest (wild-type MW of 57.7 kDa), respectively. For TRPC4 and TRPM1, an additional weak fluorescent band of lower molecular mass was observed, indicating marginal protein degradation. TRPM8 migrated as two fluorescent bands, reflecting monomeric and higher oligomeric (here dimeric) forms. In addition, the appearance of an additional blurred and smeary band for TRPM8 may indicate posttranslational glycosylation as previously reported for this protein [50]. However, as we achieved acceptable expression levels for this target, we did not attempt further optimization by removing glycosylation sites that were previously reported to enhance protein secretion [51].

Based on these highly promising production levels and indicative sample quality, as well as the fact that *S. cerevisiae*-based heterologous expression of well-behaving TRPV1 channel has been reported [52], we decided to narrow our target portfolio to characterize further one selected member from each subfamily, i.e., TRPC4, TRPV3, TRPML2 and TRPM8. 

We then evaluated the localization of the produced human TRP channels in *S. cerevisiae* (Figure 3C). As seen from live cell bioimaging fluorescent micrographs, TRPC4 and TRPML2 accumulate mainly in the plasma membrane, whereas TRPV3 and TRPM8 reside in intra-cellular compartments. Such target-specific compartmental differences in the protein accumulation are frequently observed and do not affect the final quality of the produced samples [53].

### 3.3. Solubilization Screen and F-SEC

Successful chromatography purification of membrane proteins necessitates detergent(s) to enable membrane extraction and stabilization in solution [54]. To identify the most suitable detergents for solubilization of TRPC4, TRPV3, TRPML2 and TRPM8, we assessed several non-ionic (typically milder) and zwitterionic (harsher) surfactants, but the majority displayed only partial extraction of the targets (Appendix A). Overall, non-ionic DDM and DM (tested at a concentration of 2% *w*/*w*), and zwitterionic FC-12 and FC-16 (1%) were the most efficient (Figure 4A), with the latter being able to extract 4 TRP targets with at least 44% efficacy as observed for TRPML2. Interestingly, although used at higher concentration, milder non-ionic DDM and DM were less effective, but still provided fair solubilization, with only TRPM8 channel being extracted at lower, i.e., 20% efficacy, whereas the yield for both TRPC4 and TRPV3 exceeded 40%. However, TRPML2 resisted solubilization with non-ionic detergents and was only extracted with FC-12 and FC-16.

As already mentioned, an ideal extraction detergent should also maintain protein stability, an essential factor for obtaining high-quality samples for most biochemical, biophysical and biomedical applications. To evaluate protein behavior following exposure to detergent, we again took advantage of the TEV-GFP-His8-tag and conducted F-SEC (Figure 4B–E) in presence of 0.03% DDM to minimize aggregation of tested samples, where the SEC profile of the resulting solubilized fusion protein was monitored by the fluorescence spectroscopy [55]. For each target two F-SEC runs were performed to compare the profiles after solubilization with different detergents. In case of FC-16 all obtained F-SEC profiles were sharp and symmetrical, suggesting that the TRP channels were monodispersed in this detergent. Only for TRPC4 (Figure 4B) a small fraction of the protein eluted in the void volume of the column, indicating partial aggregation. TRPC4 exhibited almost identical behavior in DDM (Figure 4B), whereas the F-SEC profile for DDM-solubilized TRPM8 was broader, but devoid of aggregation (Figure 4E). DM solubilization of TRPV3 revealed marginal aggregation and a main peak elution across a relatively large volume span, suggesting partial disruption of the tetramer (Figure 4C). The F-SEC profile of FC-12-solubilized TRPML2 demonstrated two major fluorescence peaks, indicating dissociation of the oligomeric state towards a lower oligomeric (or monomeric) state (Figure 4D).

Considering the findings from the solubilization screen and F-SEC analysis, as well as previous reports demonstrating that fos-cholines can be detrimental to membrane proteins [55,56], and the fact that both DDM and DM are popular detergents for structural and functional studies (also of TRP channels) [9,57], we decided to use primarily these milder surfactants for the subsequent large-scale purifications.

### 3.4. Purification

Affinity purification of the selected 4 targets was performed based on crude membranes isolated from 40 g fermenter-grown *S. cerevisiae* cells (obtained from ~3 L of cell culture) induced for 96 h at 15 °C (Figure 5A–D). Membranes were solubilized for 3 h at 4 °C with the detergents indicated in the respective panels of Figure 4, based on the prior F-SEC analysis. As observed from the respective chromatograms, all TRP channels eluted at imidazole concentrations of ~250 mM. The total yield of the purified protein ranged from 0.5 to 1.5 mg per liter cell culture (for TRPM8 and TRPV3, respectively) and the overall purity already following the initial purification step was fair (see insets with Coomassie-stained SDS-PAGE gels). We subsequently concentrated the top fractions and attempted TEV protease facilitated removal of the GFP-His8-tag. However, only the TRPC4- and TRPV3-fusions were successfully cleaved, whereas TRPML2 and TRPM8 remained refractory to the TEV treatment (data not shown). This likely reflects poor accessibility of the C-termini in the two latter constructs, preventing proteolytic activity. For TRPC4 and TRPV3, we performed reverse affinity purification (R-IMAC), and obtained samples of higher purity as contaminants rebound to the resin (data not shown).

The (reverse) affinity-purified TRP channels were subjected to SEC run with 3 × CMC of the corresponding detergents used for solubilization, to further verify the sample quality (Figure 6A–D). Tag-free TRPC4 eluted as a symmetric peak, indicating a high degree of homogeneity (Figure 6A). Similarly, although the SEC analysis of TRPV3 displayed some signs of aggregation, the overall shape and proportion suggested mono-dispersity of the sample (Figure 6B). We also obtained an acceptable SEC profile for TRPML2, however, the main peak was preceded by a significant shoulder, indicating the presence of higher oligomers (or even aggregation) in the sample (Figure 6C). The generated SEC profile for TRPM8 was symmetric overall, hinting again at a high homogeneity of the purified protein (Figure 6D).

### 3.5. Single Channel Current Recordings of DDM-Solubilized TRPM8

Considering the relatively high predicted molecular weight, possible glycosylation, the lowest extraction efficacy in milder detergents, hence the obtained protein yield, as well as its high medical relevance, we nominated TRPM8 channel for functional characterization efforts. Current amplitude was determined (Figure 7A,B) and plotted as a function of voltage (Vm) and a linear regression was made to determine slope conductance, as shown in Figure 7C. The single channel conductance of purified TRPM8 channels was determined to 56.9 ± 3.2 pS. 

## 4. Discussion

Membrane proteins, especially those of human origin, remain poorly understood relative to their soluble counterparts and are underrepresented in protein structure databases, largely due to difficulties in generation of prime-quality samples required for downstream efforts [58]. Furthermore, membrane proteins represent very attractive drug targets, and thus, cheap, high-efficiency, easy-to-handle and reproducible heterologous expression platforms capable of delivering large yields of recombinant human membrane proteins are still highly demanded.

The most widely exploited system for production of TRP channels from higher organisms is undoubtedly mammalian cell-based platforms that have provided several targets, including TRPV1 [14], TRPV6 [19], TRPC3 [20], TRPC4 [22], TRPM2 [27], TRPM4 [28], TRPM8 [30], TRPML1 [25], TRPA [11] and TRPPP2 [12]. However, complex growth requirements, high costs and modest yields leave room for alternative approaches [35]. Nevertheless, hitherto only two of the available structures of mammalian TRP channels, but none of human origin, originate from the protein produced in *S. cerevisiae* [15,18]. Compared to the mammalian expression systems, *S. cerevisiae* offers higher proteins yields, simpler culture conditions and significantly lower costs [59]. Importantly, as yeast can commonly contaminate mammalian cell cultures, implementation of careful aseptic routines is, however, necessary in laboratories culturing concomitantly these two types of cells [60].

In this study, we present a novel approach towards overproduction of human TRP channels employing an *S. cerevisiae*-based heterologous expression system. This platform was already successfully used for generation of several classes of membrane proteins, including the human ether-à-go-go-related (ERG) channel [49] and aquaporins [61], and even sample that yielded the first crystal structure of a human aquaglyceroporin [62]. Here, we approached the production of the challenging human TRP channels in a systematic way, starting from design of suitable plasmids, yield assessments, localization tests and solubilization screens, to affinity-based purification, SEC-based analysis and functional characterization of one selected target.

The first critical step in the optimization of recombinant protein overproduction is to assemble suitable expression plasmids [63]. Here, our strategy was based on the employment of full-length human TRP channel sequences that were codon-optimized for *S. cerevisiae*, a common practice used to improve production levels [64]. Moreover, all constructs possessed C-terminal TEV protease-detachable GFP-His8-tags, enabling detection and affinity purification of the generated targets to facilitate downstream characterization [48,65]. Furthermore, the expression vector used here encompassed several important features, including the possibility to increase the plasmid copy number and an inducible promoter [42]. Importantly, our strategy involved also the employment of a modified *S. cerevisiae* strain that overexpresses the Gal4 transcriptional activator required for galactose induced transcription [66]. Finally, we took advantage of the ability of *S. cerevisiae* to perform homologous recombination to assemble the final constructs [67], a significant economical aspect of utilizing yeast.

The initial expression tests (Figure 3A,B) already indicated the power of our production strategy, as we were able to recover all 11 human TRP channels selected for this study, all with promising production yields reaching maximum 5.2 mg of accumulated protein per liter cell culture (Figure 3A). Moreover, as evident from the in-gel GFP fluorescence signal, all targets were synthesized as membrane-embedded full-length channels with only marginal signs of protein degradation (Figure 3B), yet another important advantage of the utilized yeast platform. Moreover, for one target, namely TRPM8, we observed indications of protein glycosylation (Figure 3B). It has been previously reported that N-glycosylation is important for the function and regulation of TRP channels, including TRPA [68], TRPC [69] and TRPM [70,71] subfamilies, respectively. *S. cerevisiae* PAP1500 strain utilized here is capable of glycosylating the heterologously expressed human membrane proteins [49,61]. Of importance, *S. cerevisiae* has been shown to hyper-glycosylate proteins [36] with a detrimental impact on expression levels [37,51]. Thus, additional optimization of employed constructs by engineering glycosylation site(s) could be beneficial to further enhance the TRP channel production yields and activity, however, it was not attempted here.

The initial expression tests (Figure 3A,B) already indicated the power of our production strategy, as we were able to recover all 11 human TRP channels selected for this study, all with promising production yields reaching maximum 5.2 mg of accumulated protein per liter cell culture (Figure 3A). Moreover, as evident from the in-gel GFP fluorescence signal, all targets were synthesized as membrane-embedded full-length channels with only marginal signs of protein degradation (Figure 3B), yet another important advantage of the utilized yeast platform. Moreover, for one target, namely TRPM8, we observed indications of protein glycosylation (Figure 3B). It has been previously reported that N-glycosylation is important for the function and regulation of TRP channels, including TRPA [68], TRPC [69] and TRPM [70,71] subfamilies, respectively. *S. cerevisiae* PAP1500 strain utilized here is capable of glycosylating the heterologously expressed human membrane proteins [49,61]. Of importance, *S. cerevisiae* has been shown to hyper-glycosylate proteins [36] with a detrimental impact on expression levels [37,51]. Thus, additional optimization of employed constructs by engineering glycosylation site(s) could be beneficial to further enhance the TRP channel production yields and activity, however, it was not attempted here.

Mammalian TRP channels are known to differ in subcellular localization and, in addition to the plasma membrane, can localize to intracellular membranes [72]. For four selected TRP channels, we also observed differences in localization in the *S. cerevisiae* cells (Figure 3C), with TRPV3 and TRPM8 accumulating mainly intracellularly. In accordance, intracellular membrane localization has previously been observed for these two targets also in human cells [72,73].

To enable successful TRP channel production, suitable detergents for membrane extraction and maintaining protein stability were identified. The initial choice of detergents for solubilization was based on our previous large-scale analysis, which compared the efficacy of different compounds for extracting overproduced human aquaporins using the same expression platform [61]. In the solubilization screen presented here, we discovered that harsher zwitterionic detergents of the fos-choline family displayed stronger extraction properties of human TRP channels from the *S. cerevisiae* membranes as compared to milder non-ionic maltosides (Figure 4A, Appendix A). Indeed, similar observations have also been reported for other membrane protein classes overexpressed in the same host [74]. In general, all 4 targets tested here were rather resistant to solubilization, with only TRPM8 channel demonstrating extraction efficacy in FC-16 exceeding 80%. Moreover, when using FC-16, we observed slightly higher solubilization efficacy of the two targets accumulating in the intracellular membranes (i.e., TRPV3 and TRPM8) as compared to TRPC3 and TRPML2 that localized to the plasma membrane. A similar trend was previously reported for the human aquaporin family produced using the same platform [61]. Fos-choline-solubilized material exhibited moderately more symmetric F-SEC profiles, however, the mono-dispersity of the maltoside-extracted material was also relatively high (Figure 4B–E), indicating that milder detergents are promising for isolation of human TRP channels from crude *S. cerevisiae* membranes. This is also in agreement with the two above-mentioned structural reports of human TRPV channels that were overproduced in *S. cerevisiae* cells and solubilized in maltosides [15,18].

Affinity-based protein purification yielded milligram amounts of relatively pure samples already following a single chromatography round (Figure 5A–D). Removal of GFP-His8-tag by TEV protease cleavage followed by reverse affinity purification was successful for two out of four targets. However, optimization of neither sequences of employed constructs nor cleavage conditions were attempted here. The quality of (reverse) affinity-purified samples was evaluated by SEC performed with the primary detergent, i.e., the respective detergent used for solubilization and affinity purification for each target, respectively. Strikingly, all tested protein samples, including human TRPML2 isolated using harsh FC-16, exhibited encouraging SEC profiles (Figure 6A–D).

In our final quality control test, we assessed whether human TRP channels produced using the described overproduction approach preserved functionality, an important checkpoint for any heterologous expression platform. DDM-solubilized human TRPM8 was selected as a representative medically relevant target, with the aim to compare its conductance with permeation properties of TRPM8 isolated from other hosts. The slope conductance of TRPM8 reconstituted in lipid bilayers was 56.9 ± 3.2 pS (Figure 7C). Similar conductance (55 ± 3.9 pS) was found for TRPM8 from *R. norvegicus* expressed in *Xenopus laevis* oocytes [75]. Moreover, a study employing human kidney cells reported mean TRPM8 conductance of 72 ± 12 pS for outward currents, 42 ± 6 pS for inward currents, respectively, and sub-conductance burst openings at 30 ± 3 pS, suggesting that the channel has a strong outward rectification [76]. Thus, our functional data obtained for human TRPM8 isolated from *S. cerevisiae* are in accordance with the conductance of both rat and human channels expressed in higher eukaryotic heterologous systems, with subtle quantitative differences that likely are attributed to variations of the experimental conditions.

All-in-all, obtained results establish our *S. cerevisiae*-based platform as a cost-effective and high-throughput approach to produce milligram quantities of stable and functional human TRP channels. Thus, the approach can be applied in functional, pharmacological and structural studies of this important family of ion-conducting proteins.

## Figures and Tables

**Figure 1 cells-08-00148-f001:**
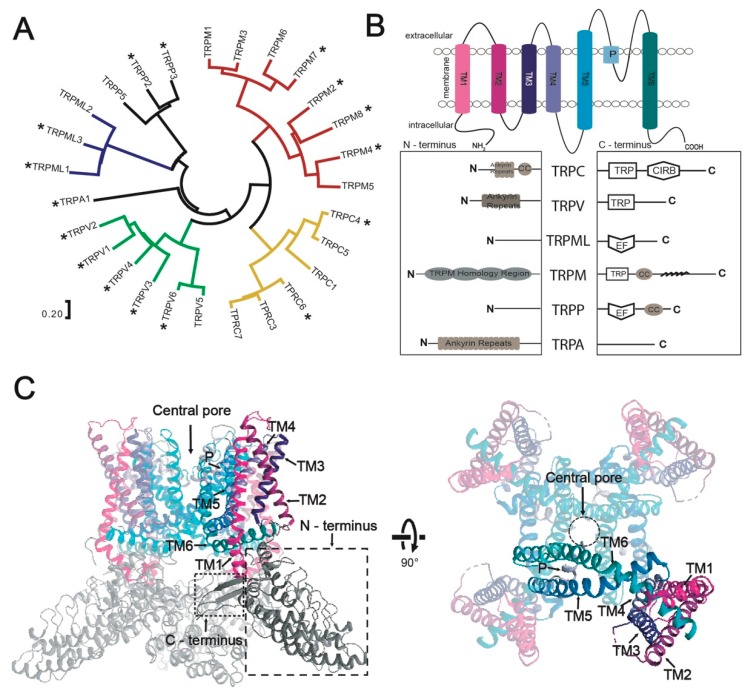
Human transient receptor potential (TRP) channels. (**A**) Phylogenetic distribution of the human TRP channel family including six subfamilies comprising proteins with distinct channel properties. Protein sequences were aligned using MEGA7 (https://www.megasoftware.net/). Structurally determined channels are highlighted with stars (not all structures were of protein with human origin, see also Table 1). (**B**) Topology of TRP channels showing in detail distinct architecture of the intracellular N- and C-termini across TRP channel subfamilies. (**C**) Overall structure of TRP channels with three out of the four monomers shown in pale colors (the structure of TRPV1, PDB-ID: 3J5P [14], was used as a model). TRP channel monomers consist of six transmembrane helices (TM1 to TM6) that assemble as tetramers with a single ion conducting central pore in the center formed by TM5, TM6 and the interconnecting pore-loop (P).

**Figure 2 cells-08-00148-f002:**
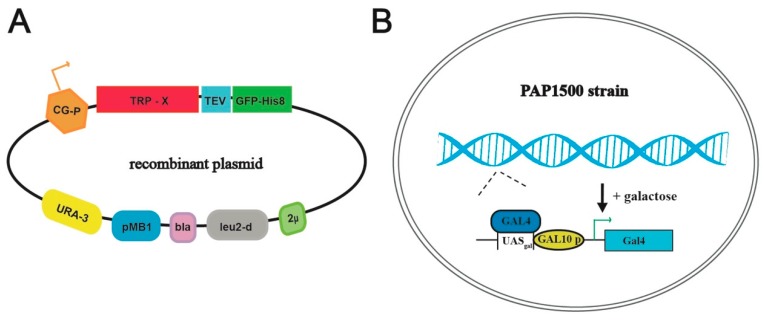
Schematic overview of the yeast production system. (**A**) Map of the employed plasmid encoding the respective TRP channels fused C-terminally with a Tobacco Etch Virus, TEV, protease cleavage site followed by green fluorescent protein sequence, GFP, attached to an octa-histidine stretch, His8-tag (TEV-GFP-His8-tag). Other critical elements of the plasmid include 2μ (yeast 2 micron origin of replication), leu2-d (poorly expressed allele of the β-iso-propyl-malate dehydrogenase gene), bla (β-lactamase gene), pMB1 (pMB1origin of replication), URA3 (yeast orotidine-5’-phosphate decarboxylase gene) and CG-P (hybrid promoter of GAL10 upstream activating sequence and 5’ non-translated leader of cytochrome-1 gene). (**B**) The *Saccharomyces cerevisiae* protein production strain PAP1500 overexpresses the Gal4 transcriptional activator in the presence of galactose. Gal4 is the limiting factor for expression from galactose regulated promoter. GAL10 p, UASgal, a specific DNA binding site for GAL4 activator.

**Figure 3 cells-08-00148-f003:**
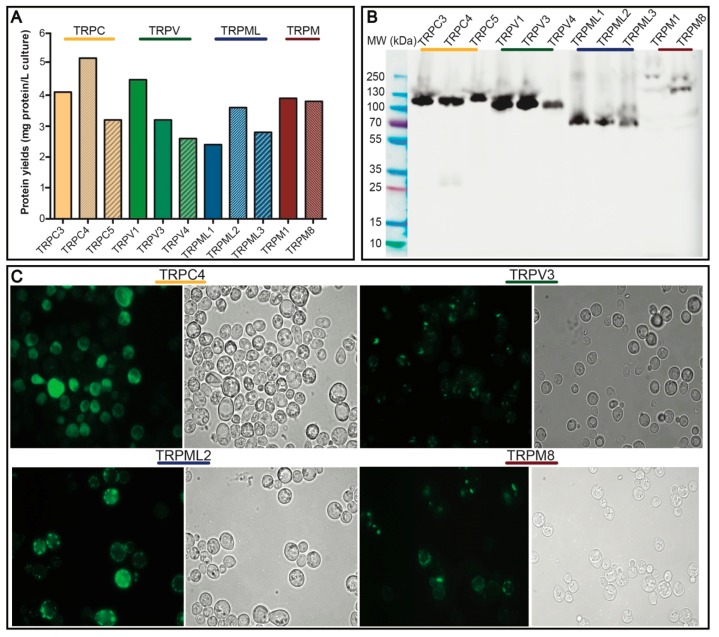
Expression and localization of human TRP channels. Data are shown for *S. cerevisiae* cell cultures grown in 2-L scale for 48 h at 15 °C. (**A**) Estimates of protein production levels of 11 human TRP-channels. Measured whole-cell GFP fluorescence was converted to the protein amount and the predicted yield is shown as mg per liter cell of culture. (**B**) In-gel GFP fluorescence of SDS-PAGE-separated crude membranes containing the different TRP channel targets. The image includes all samples separated on the same gel. (**C**) Live cell bioimaging of yeast cells expressing selected TRP channels. For each target GFP fluorescence and differential interference contrast micrographs are shown. Magnification: 1000×.

**Figure 4 cells-08-00148-f004:**
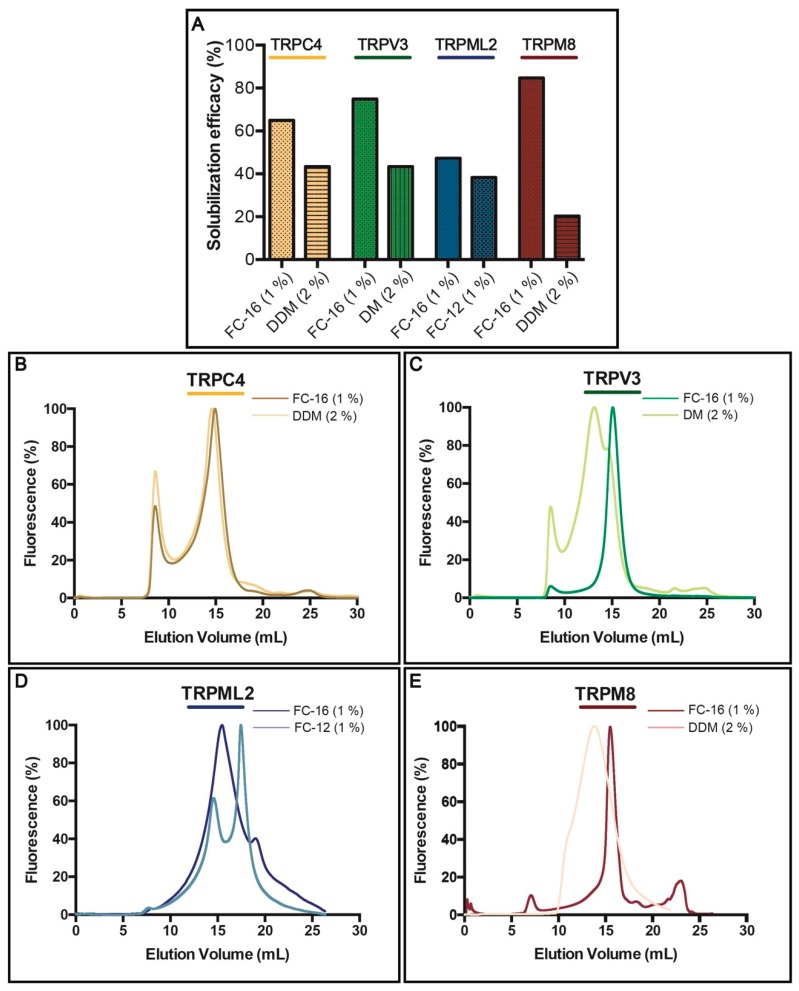
Solubilization and fluorescence-detection size-exclusion chromatography (F-SEC) of selected human TRP channels. (**A**) Solubilization efficacy in FC-12 (at a concentration of 1%), FC-16 (1%), DDM (2%) and DM (2%) of crude *S. cerevisiae* membranes for selected TRP channels. Solubilization was performed for 2 h at 4 °C and GFP fluorescence of the solubilized material (supernatant following ultracentrifugation) was used to calculate the percentage of extraction. (**B**–**E**) F-SEC analysis of the detergent-solubilized samples. Concentrations of the corresponding detergents used during solubilization are indicated. Normalized F-SEC profiles were obtained for solubilized material separated on Superose 6 HR 10/30 column where GFP fluorescence was monitored. The void volume of the column (Superose 6 HR 10/30) is ~8 mL.

**Figure 5 cells-08-00148-f005:**
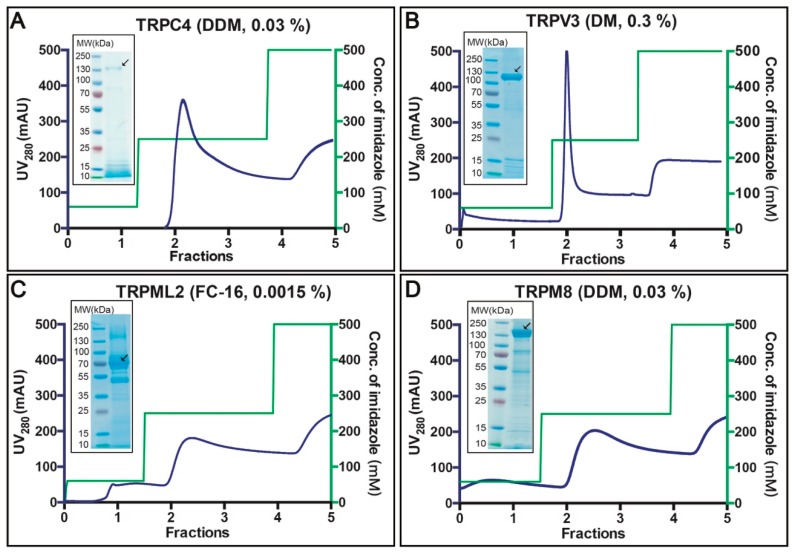
Purification of selected human TRP channels. (**A**–**D**) Selected targets were purified from crude *S. cerevisiae* membranes isolated from 40 g of fermenter-grown cells (obtained from ~3 L of cell culture) induced for 96 h at 15 °C. Membranes were solubilized with the indicated detergents for 3 h at 4 °C. Affinity protein purification was performed using immobilized metal ion affinity chromatography (IMAC). Concentrations of the corresponding detergents used for protein elution are indicated. IMAC profiles display UV_280_ signal for eluted protein (blue) and the corresponding imidazole gradients used for elution (green). Insets: Coomassie-stained SDS-PAGE gels with concentrated affinity-purified samples. Black arrows indicate the MW for the corresponding fusion protein.

**Figure 6 cells-08-00148-f006:**
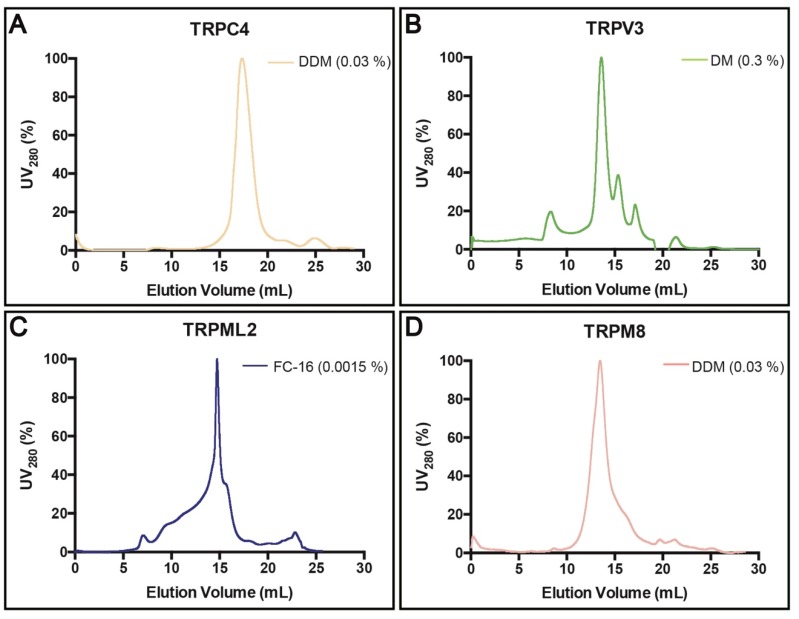
Size-exclusion chromatography (SEC) of selected human TRP channels. (**A**–**D**) Normalized SEC profiles for protein samples previously purified using reverse (TRPC4, A and TRPV3, B) or direct (TRPML2, C and TRPM8, D) affinity chromatography (immobilized metal ion affinity chromatography, IMAC). Concentrations of the corresponding detergents used during SEC are indicated. The void volume of the column (Superose 6 HR 10/30) is ~8 mL.

**Figure 7 cells-08-00148-f007:**
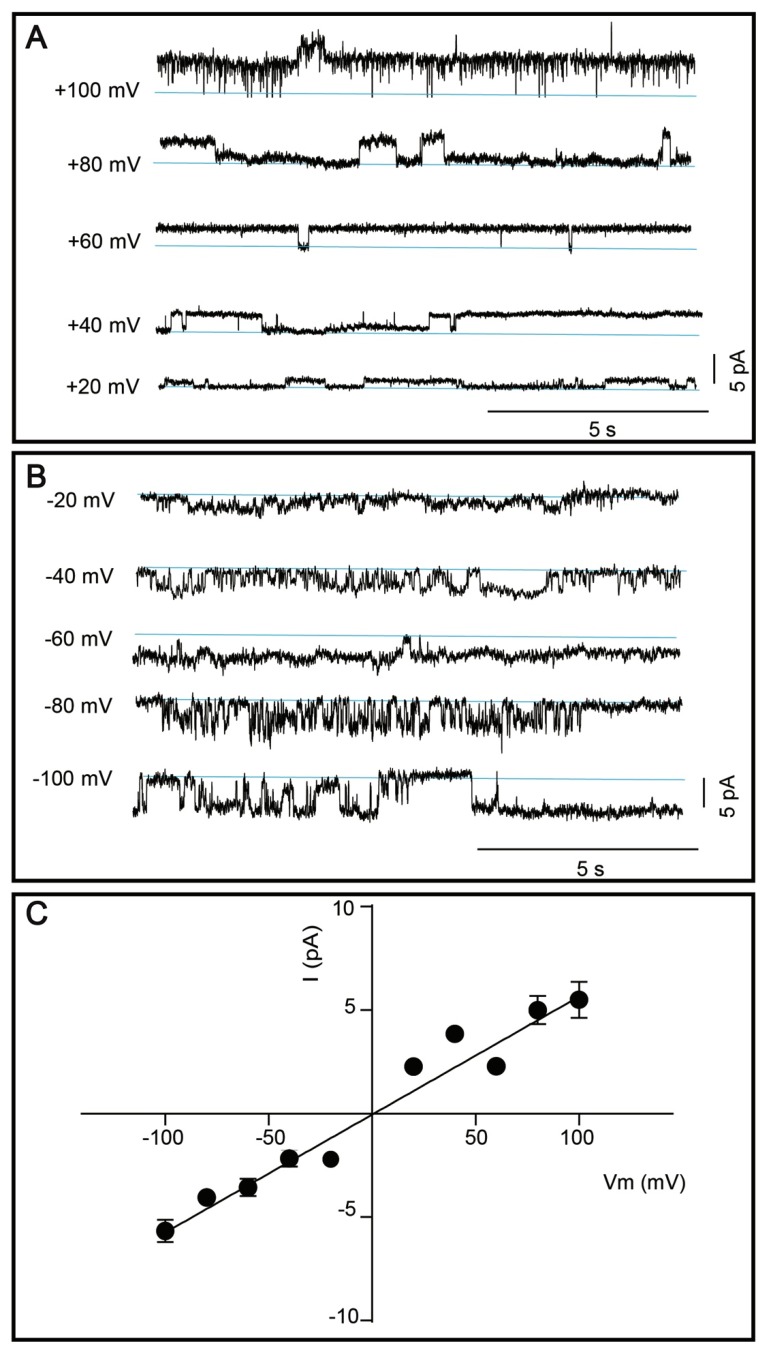
Single channel activity of affinity and size-exclusion chromatography purified TRPM8. (**A**–**B**) Representative single channel currents of TRPM8 reconstituted in the planar lipid bilayer membranes. Currents were elicited by positive (**A**) and negative (**B**) voltage steps. Channel activity is observed at all applied voltages. Zero current is indicated by the blue lines. (**C**) Current (I) was plotted as a function of voltage (V) and a linear regression was fitted to the data to determine the slope conductance (56.9 ± 3.2 pS). Results are shown as mean ± SEM, each data point was based on 2–52 single channel events.

**Table 1 cells-08-00148-t001:** Structurally determined TRP channels. TRP channels produced in *Saccharomyces cerevisiae* are framed. Targets included in this study are indicated in colors (yellow, green, blue and red represent TRPC, TRPV, TRPML and TRPM subfamilies, respectively). Boxes indicate the recombinant proteins produced in yeast, targets purified in this study are shown in bold. Additional information about constructs, structure determination method and obtained resolution are also shown.

Target	Organism	Expression System	Full Length	Truncation	Structure Determination Method	Resolution Å
TRPC3	Homo sapiens	Homo sapiens (AD-HEK293)	x		cryo-EM	4.4
**TRPC4**	Mus musculus	Spodoptera frugiperda		x	cryo-EM	3.3
TRPC5	Mus musculus	Homo sapiens (HEK293S)		x	cryo-EM	2.9
TRPC6	Mus musculus	Spodoptera frugiperda		x	cryo-EM	3.8
TRPC6	Homo sapiens	Homo sapiens (AD-HEK293)	x		cryo-EM	3.8
TRPV1	Rattus norvegicus	Homo sapiens (HEK293S)		minimal function	cryo-EM	4.2
TRPV2	Rattus norvegicus	Saccharomyces cerevisiae	x		cryo-EM	4.4
TRPV2	Oryctolagus cuniculus	Spodoptera frugiperda	x		cryo-EM	3.8
TRPV2	Oryctolagus cuniculus	Spodoptera frugiperda		minimal function	X-ray	3.9
**TRPV3**	Mus musculus	Homo sapiens (HEK293S)	x		cryo-EM	4.3
TRPV4	Xenopus tropicalis	Pichia pastoris		x	cryo-EM	3.8
TRPV5	Oryctolagus cuniculus	Saccharomyces cerevisiae	x		cryo-EM	3.9
TRPV6	Rattus norvegicus	Homo sapiens (HEK293S)		x	X-ray	3.2
TRPV6	Homo sapiens	Homo sapiens (HEK293S)	x		cryo-EM	3.6
TRPML1	Homo sapiens	Homo sapiens (HEK293S)	x		cryo-EM	3.7
TRPML3	Callithrix jacchus	Spodoptera frugiperda	x		cryo-EM	2.9
TRPM2	Danio rerio	Homo sapiens (HEK293S)	x		cryo-EM	3.3
TRPM4	Homo sapiens	Homo sapiens (HEK293F)	x		cryo-EM	3.2
TRPM7	Mus musculus	Homo sapiens (HEK293S)		x	cryo-EM	3.3
**TRPM8**	Ficedula albicollis	Homo sapiens (HEK293S)	x		cryo-EM	4.1
TRPA1	Homo sapiens	Homo sapiens (HEK293)	x		cryo-EM	4.2
TRPP2	Homo sapiens	Homo sapiens (HEK293S)		x	cryo-EM	4.2
TRPP3	Homo sapiens	Homo sapiens (HEK293S)	x		cryo-EM	3.1

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
