# Peer review of "Purification of Functional Human TRP Channels Recombinantly Produced in Yeast"

_cells, 2019, doi:10.3390/cells8020148_

Round 1

Reviewer 1 Report

The manuscript by Zhang et al. reports the successful production of 11 different TRP channels in yeast. Four TRP proteins were solubilized and purified and one (TRPM8) channel was functional characterized by single channel current recordings in planar lipid bilayers. This an excellent manuscript demonstrating the great potential of production of TRP proteins in yeast. I have only minor comments:

(1)    In my opinion the title is misleading. You are not able to isolate human channels from native yeast cells. Please change to: “Recombinant production of human TRP channels in yeast: purification, solubilization and functional characterization. “ or similar.

(2)    Reference 8 is too old to inform on currently running clinical trials on TRP channels.

(3)    Figure 1 is too small. It would be better to show two larger figures; one  describing TRP channels and the other one with the cloning strategy.

(4)    Table 1: A reference is missing on production of murine TRPC6 in SF-9 cells (JBC 293(26)10381–10391 (2018)). Please specify the expression system “homo sapiens”. Which human cells were used?

(5)    Please specify codon optimization for S. cerevisiae. Which codons were changed?

(6)    Protein glycosylation is an important issue for production in S. cerevisiae. Next to TRPM8, which is mentioned in the text, TRPC3 is a target protein for glycosylation. Glycosylation may influence channel activity (see JBC 278 (48) 47842–47852 (2003)), but may also  decrease efficient protein production. Therefore, some  researchers delete glycosylation sites in TRP cDNAs for a more efficient production. How good are the chances for a correct glycosylation of TRP channels in yeast and does it inhibit protein yields? If you don’t have own data, please at least discuss this important point.

(7)    Although yeast seems to be a good expression system, the risk of contamination of mammalian cell cultures in labs working with eukaryotic cells is high.  This point should be mentioned.

Author Response

 Point-to-point response to the Reviewers' comments

We thank the Editor for the positive evaluation of our paper as well as both Reviewers for recognizing the merit of our work and for providing very valuable critique. As detailed below, we have addressed the comments of Reviewers #1 and #2 fully and carefully as requested by the editor, and revised the manuscript accordingly. To facilitate the reviewing process, we have copied the Reviewers’ original comments, which are shown in black. Our responses are shown in green. Any amendments to the reviewed version of our manuscript are visible in the revised, i.e., “track-changes” version of the paper.

General comment

We have revised the original reference list and included the most recently published papers as well as some references suggested by the Reviewers.

Reviewer’s #1 comments:

The manuscript by Zhang et al. reports the successful production of 11 different TRP channels in yeast. Four TRP proteins were solubilized and purified and one (TRPM8) channel was functional characterized by single channel current recordings in planar lipid bilayers. This an excellent manuscript demonstrating the great potential of production of TRP proteins in yeast. I have only minor comments:

We thank the reviewer for the work and we are pleased to see that we share the opinion that our manuscript demonstrates the great potential of TRP channels production in yeast for downstream applications.

        (1)    In my opinion the title is misleading. You are not able to isolate human channels from native yeast cells. Please change to: “Recombinant production of human TRP channels in yeast: purification, solubilization and functional characterization. “or similar.

We agree with the reviewer that the previous title was not sufficiently precise. Therefore, following the suggestion, we changed the title to “Recombinant production of functional human TRP channels in yeast”. The new title indicates directly that the TRP channel production is performed using yeast-based expression system, but not yeast cells as a native source. 

        (2)    Reference 8 is too old to inform on currently running clinical trials on TRP channels.

We agree with the reviewer and exchanged the above-mentioned reference to the more recent one (from 2018): Moran, M.M. TRP Channels as Potential Drug Targets. Annu. Rev. Pharmacol. Toxicol. 2018, 58, 309–330, doi:10.1146/annurev-pharmtox-010617-052832.

        (3)    Figure 1 is too small. It would be better to show two larger figures; one describing TRP channels and the other one with the cloning strategy.

We thank the reviewer for this suggestion. Accordingly, we split Fig. 1 into two figures: new Fig. 1 showing panels A-C from the previous version and new Fig. 2 presenting the overview of yeast production system (now as panels A and B), respectively. The numbering of the subsequent figures was changed consequently.

        (4)    Table 1: A reference is missing on production of murine TRPC6 in SF-9 cells (JBC 293(26)10381–10391 (2018)). Please specify the expression system “homo sapiens”. Which human cells were used?

We thank the reviewer for pointing this out. The missing piece of information about the murine TRPC6 structure was incorporated into the Table 1 (with the corresponding reference in the main text). We also specified which human cell lines were used for the expression of the respective TRP channels. Moreover, when revising the Table 1, we corrected two errors that were present in its previous version. 

        (5)    Please specify codon optimization for S. cerevisiae. Which codons were changed?

As mentioned in the Materials and Methods section, we purchased codon-optimized cDNA from GenScript. Their “OptimumGene™ PSO” algorithm takes into consideration a variety of critical factors involved in different stages of protein expression, such as codon adaptability, mRNA structure and various cis-elements in transcription and translation. This information is now incorporated into the above-mentioned section.

        (6)    Protein glycosylation is an important issue for production in S. cerevisiae. Next to TRPM8, which is mentioned in the text, TRPC3 is a target protein for glycosylation. Glycosylation may influence channel activity (see JBC 278 (48) 47842–47852 (2003)), but may also decrease efficient protein production. Therefore, some researchers delete glycosylation sites in TRP cDNAs for a more efficient production. How good are the chances for a correct glycosylation of TRP channels in yeast and does it inhibit protein yields? If you don’t have own data, please at least discuss this important point.

We thank the reviewer for raising this important issue. As we did not attempt engineering glycosylation in the present work, we discussed the importance of glycosylation in regard to both TRP channels and, overall, recombinant protein expression. Hence, we updated Results and Discussion sections accordingly.

        (7)    Although yeast seems to be a good expression system, the risk of contamination of mammalian cell cultures in labs working with eukaryotic cells is high.  This point should be mentioned.

We definitely agree with the reviewer and, hence, we now mention this important issue in Discussion section.

Reviewer 2 Report

A manuscript of Zhang et al reports a new approach for isolation of functional TRP channels expressed in the yeast Saccharomyces cerevisiae. The manuscript is well written, the main conclusions are supported by original experimental data. The results are new and may have a significant impact on the field. However, there are some critical points which should be addressed prior to acceptance:

 1. Table 1 indicates that the cryo-EM TRPM7 structure was obtained for the full-length TRPM7 protein. In fact, a truncated variant of TRPM7 was examined. Please, correct.

 2. It is not well explained why particular 11 TRP channels were selected for the study and why TRPM8 was nominated for a functional assessment in lipid bilayers.

3. A functional assessment of single channel activity of TRPM8 in lipid bilayers is the most valuable information in the manuscript. However, these experiments incompletely described by the authors. For instance, it was not reported whether TRPM8 was active only in the presence of an agonist and whether TRPM8 was sensitive to temperature and PIP2. These controls are crucial for the claim that the functional TRPM8 channel was successfully produced.

Author Response

 Point-to-point response to the Reviewers' comments

We thank the Editor for the positive evaluation of our paper as well as both Reviewers for recognizing the merit of our work and for providing very valuable critique. As detailed below, we have addressed the comments of Reviewers #1 and #2 fully and carefully as requested by the editor, and revised the manuscript accordingly. To facilitate the reviewing process, we have copied the Reviewers’ original comments, which are shown in black. Our responses are shown in green. Any amendments to the reviewed version of our manuscript are visible in the revised, i.e., “track-changes” version of the paper.

General comment

We have revised the original reference list and included the most recently published papers as well as some references suggested by the Reviewers.

Reviewer’s #2 comments:

A manuscript of Zhang et al reports a new approach for isolation of functional TRP channels expressed in the yeast Saccharomyces cerevisiae. The manuscript is well written, the main conclusions are supported by original experimental data. The results are new and may have a significant impact on the field. However, there are some critical points which should be addressed prior to acceptance:

We thank the reviewer for the work and we are pleased to see that we share the opinion that our manuscript includes results that may have a significant impact on the field.

1.     Table 1 indicates that the cryo-EM TRPM7 structure was obtained for the full-length TRPM7 protein. In fact, a truncated variant of TRPM7 was examined. Please, correct.

We thank the reviewer for finding this mistake that is now corrected in the revised version of Table 1.

2.     It is not well explained why particular 11 TRP channels were selected for the study and why TRPM8 was nominated for a functional assessment in lipid bilayers.

We agree with the reviewer that our rationalization of the target selection was insufficiently explained. We aimed at selection of TRP channels representing main subfamilies, thus possessing a variety of distinct structural and functional features. Naturally, our selection was limited by the number of readily available expression constructs that could be included in the study to avoid unnecessary delays. As far as the nomination of TRPM8 for the functional characterization is concerned, our reasoning was multivariate. Biochemically, TRPM8 possesses relatively high predicted molecular weight, it is possibly glycosylated and proves to have the lowest extraction efficacy in milder detergents. Hence the obtained protein yield of all tested targets was the smallest for TRPM8. Medically, TRPM8 is an important pharmacological target proposed in treatment of both cold-associated respiratory disorders and prostate cancer.

Hence, considering the reviewer’s suggestion, we now improved explanation of our reasoning through the manuscript.

    3. A functional assessment of single channel activity of TRPM8 in lipid bilayers is the most valuable information in the manuscript. However, these experiments incompletely described by the authors. For instance, it was not reported whether TRPM8 was active only in the presence of an agonist and whether TRPM8 was sensitive to temperature and PIP2. These controls are crucial for the claim that the functional TRPM8 channel was successfully produced.

Production of human membrane proteins in yeast is challenging. The scope of the present manuscript was to demonstrate that functional human TRP channels can be produced in yeast and to describe the method we have established. The functional assessment of TRPM8 was included to show that functional activity is maintained, suggesting that the proteins are properly folding. We agree with the reviewer that it would be interesting to perform a detailed characterization of TRPM8, including the effects of temperature and PIP2, however, it would be a paper in its own right.

We performed initial experiment in the absence of icilin, however, channel activity was sparse, but not absent. As our aim mainly was to detect channel openings and determine slope conductance, we opted to add icilin in the following experiments.

Round 2

Reviewer 2 Report

The authors have addressed all my concerns.